# MRI May Be More Valuable than Pelvic Radiographs in the Assessment of Paediatric Borderline Acetabular Dysplasia

**DOI:** 10.3390/children10040758

**Published:** 2023-04-21

**Authors:** Hans-Christen Husum, Michel Bach Hellfritzsch, Mads Henriksen, Martin Gottliebsen, Ole Rahbek

**Affiliations:** 1Interdisciplinary Orthopaedics, Aalborg University Hospital, 9000 Aalborg, Denmark; 2Department of Radiology, Aarhus University Hospital, 8000 Aarhus, Denmark; 3Department of Orthopaedics, Aarhus University Hospital, 8000 Aarhus, Denmark

**Keywords:** Paediatric Orthopaedics, acetabular dysplasia, Magnetic Resonance Imaging

## Abstract

The osseous acetabular index (OAI) and cartilaginous acetabular index (CAI) is often used in diagnosing acetabular dysplasia (AD) in children. We examined the reliability of OAI and CAI in AD diagnostics and compared OAI measurements obtained from radiographs versus MRI. Four raters performed retrospective repeated measurements of the OAI and CAI on pelvic radiographs and MRI scans of 16 consecutive patients (mean age 5 years (2–8)) examined for borderline AD during a period of 2½ years. In MRI, the image selected for analysis by the raters was also registered. Spearman’s correlation, scatter plots, and Bland–Altman (BA) plots were analysed for correlation between OAI on pelvic radiographs (OAIR) and MRI scans (OAIMRI), while intra- and interrater reliability was assessed for OAIR, OAIMRI, CAI, and MRI image selection using intraclass correlation coefficients (ICC). ICC values for inter- and intrarater reliability of OAIR, OAIMRI, and CAI were all above 0.65, with no significant differences observed. ICC values (CI) for individual raters’ MRI image selection was 0.99 (0.998–0.999). The mean difference (95% CI) between OAIR and OAIMRI was −0.99 degrees (−1.84; −0.16), while the mean absolute difference (95% CI) between OAIR and OAIMRI was 3.68 degrees (3.17; 4.20). Absolute differences between OAIR and OAIMRI was independent of pelvic positioning or time interval between radiographs and MRI scans. OAI and CAI had high Intrarater reliability but mediocre interrater reliability. There was an absolute difference of 3.7 degrees in OAI between pelvic radiographs and MRI scans.

## 1. Introduction

When deciding the management and prognosis of cases of acetabular dysplasia (AD), surgeons primarily use radiographic measurements, with the osseous acetabular index (OAI) as the most commonly used [1].

However, the interpretation of radiographs and their associated measurements are dependent on the positioning of the pelvis [2], which is influenced by children’s general inability to maintain their position during radiographic examination.

In a study by Omeroglu et al. [1], experienced surgeons were presented with retrospective mid-term follow-up radiographs of children having undergone initial soft tissue surgery for DDH. Surgeons were blinded to the final outcome of the hips. Forty percent of the surgeons chose not to recommend surgery on hips that turned out dysplastic at skeletal maturity and 12% of surgeons would operate on hips that would normalize with time without the need for additional treatment.

Therefore, alternative reliable prognosticators for AD in surgical decision making has been sought. The cartilaginous structures of the paediatric hip have long been recognized as important structures to consider when selecting patients for corrective AD surgery [3], and, in 2008, the cartilaginous acetabular index (CAI) was first proposed as a new measurement to be used when evaluating AD patients for surgery [4]. The CAI develops differently than its osseous counterpart and may predict osseous developmental capacity [5]; thus, it has the potential to be a prognosticator for long term outcome of hip dysplasia. Previous studies have shown significant differences when comparing OAI used in evaluating AD to their cartilaginous counterparts and suggest using the CAI as indicator for corrective surgery [6]. To our knowledge, there is no formal agreement on how to select sequences and images for OAI and CAI measurements in MRI, and the reliability of this selection has not been addressed sufficiently.

No studies have examined the reliability of OAI and CAI measured on children evaluated for acetabular dysplasia while taking pelvic positioning in radiographs and image selection in MRI into account.

The purpose of this study was to assess the inter- and intrarater reliability when measuring OAI and CAI on pelvic radiographs and MRI scans of children examined for borderline AD with no previous surgical treatment for hip dysplasia. Furthermore, we wished to analyse the differences in OAI when measured on pelvic radiographs versus MRI and their correlation to pelvic positioning and time interval between examinations. Finally, the interrater agreement in selection of MRI images for the abovementioned measurements was evaluated.

## 2. Methods

This was a retrospective cohort study of all children examined for borderline AD with both pelvic radiographs and MRI at the departments of Orthopaedics and Radiology at INSTITUTION, in the period September 2016 to April 2019.

The reporting follows STROBE and GRRAS guidelines for reporting observational and reliability studies [7,8].

We included both hips of all children evaluated for borderline AD who had pelvic MRI scans performed supplementary to pelvic radiographs with no previous surgical treatment for hip dysplasia.

The group of raters consisted of two paediatric orthopaedic surgeons (XX, XX) and two musculoskeletal radiologists (XX, XX). All raters had a minimum of seven years of experience in paediatric radiograph interpretation and five years of experience in musculoskeletal MRI interpretation, except XX, who had three months of experience in MRI interpretation.

No sample size calculation was made. This was a convenience sample based on available patients at our institution during the 2½ year period.

MRI scans were performed at INSTITUTION using five MRI scanners: Philips Medical Systems: Achieva dStream 3.0T, Ingenia 1.5T. GE Medical Systems: Optima MR 450w 1.5T. Siemens: Skyra 3T, Avanto fit 1.5T.

The scans were coronal T1-weighted spin echo images, and a transverse scout view of the acetabular region and symmetrical coronal sections was used to determine imaging sections. Slice thickness was 3 to 4 mm with the most central section chosen. The child was scanned in a supine position with a body array coil placed anteriorly and posteriorly to the hip joint. The scan time was five minutes. Pelvic radiographs were performed with the child in the supine examination position. Scans were archived and viewed using Picture Archiving and Communication Software (PACS) (Impax, client 6.5 AGFA Healthcare N.V., Mortsel, Belgium).

Measurements were performed retrospectively on T1-weighted pelvic MRI scans and pelvic radiographs. The raters measured the osseous acetabular indexes (OAI), according to the method described by Tönnis [9]. The cartilaginous acetabular index (CAI) was measured on MRI scans, according to the description by Zamzam et al. [4], as the angle between Hilgenreiner’s line and a line drawn from the lateral margin of the cartilaginous acetabulum at the attachment of the labrum to the superolateral margin of the triradiate cartilage (Figure 1).

Prior to performing measurements for this study, a pre-study consensus meeting was held to ensure agreement between raters on how measurements should be performed. The measurements were made in the following weeks at each rater’s convenience and were repeated after a minimum of one week after initial measurements. All measurements were registered and stored in an encrypted standardized personal data sheet for each rater and could not be edited once all measurements had been saved. Measurements were performed in chronological order of patients, which was not randomized for the repeat measurement exercise.

The pelvic tilt index and pelvic rotation index used for correlation analysis in this study were based on measurements made by the senior musculoskeletal radiologist (XX) and were measured according to the descriptions of Ball and Kommenda and Tönnis [9,10].

The raters were blinded to the measurements of their fellow raters, their former measurements when performing the repeat measurements, and patient information, in addition to MRI scans and pelvic radiographs. They were aware that their measurements would be compared.

No ethical approval was sought in accordance with the guidelines on non-interventional health studies from the National Committee on Health Research Ethics.

### Statistical Methods

We included bilateral measurements for all children when analysing mean values for OAI on radiographs and MRI (OAIR/OAIMRI) and CAI. A nested mixed model was used with random effects for patient ID, side, and the measurement method, as fixed effect was used to account for the bilaterality in data, and the variation across patient and side was measured. As model control, the qq-plots of the residuals were investigated.

To account for the bilaterality in data when performing inter- and intrarater reliability analysis, the ICC calculations were performed using 1000 bootstraps, each containing all right or all left measurements for a specific patient, totalling 16 measurements for each bootstrap sample. Intrarater ICC values were calculated only for the first round of rating, with an interrater ICC as a two-way, mixed effect per single rater, with absolute differences and intrarater ICCs as two-way mixed effects per single rater with absolute agreements.

Agreement in MRI image selection was calculated on patients where identical series were selected. We included image selection from the first round of rating in this analysis. ICC was calculated as two-way mixed effect per single rater with absolute agreement.

Absolute differences in measurements between radiographs and MRI scans were calculated for each rater for the first round of rating and correlated to pelvic tilt index, pelvic rotation index, and interval between scans using Spearman’s correlation. Additionally, scatter plots of absolute differences between OAIR and OAIMRI in bilateral measurements for all individuals measured by the senior radiologist (XX) in 1 round of rating were investigated for any association not detected by the Spearman correlation analysis.

Outcomes were presented as ICC values and mean values, with accompanying 95% confidence intervals.

We interpreted ICC values according to the general guidelines of Portney and Watkins [11] with an ICC value of 0 representing no reliability, 0.75 representing good reliability, and an ICC value of 1 representing complete reliability.

## 3. Results

We identified 16 children (14 girls). One case was missing a pelvic radiograph and one was missing a pelvic MRI scan. All MRI scans but one were performed without sedation. We included pelvic radiographs and MRI scans of 30 hips in total. For image agreement analysis, we further excluded six hips for lacking agreement in MRI sequence selection, leaving 24 hips for image agreement analysis. The mean age at pelvic radiograph was 5 years (range 2;8), with mean time intervals between pelvic radiographs and MRI scans being 133 days (range 16;234).

ICC values (95% CI) for interrater reliability of measurements were 0.84 (0.65; 0.93), 0.72 (0.51; 0.84), and 0.65 (0.44; 0.80) for OAIR, OAIMRI, and CAI, respectively. ICC values (95% CI) for intrarater reliability between measurements were 0.91 (0.72; 0.98), 0.84 (0.55; 0.97), and 0.80 (0.56; 0.95) for OAIR, OAIMRI, and CAI, respectively (Table 1). No significant differences were observed.

ICC values (95% CI) for interrater agreement in image selection was 0.99 (0.998; 0.999).

The mean difference (95% CI) between OAIR and OAIMRI was −1.0 degrees (−1.8;−0.2) (Figure 2), while the mean absolute difference (95% CI) between OAIR and OAIMRI was 3.7 degrees (3.2; 4.2).

There was no significant difference between OAIR and OAIMRI (Table 2). There was no significant correlation between mean absolute difference of OAIR and OAIMRI and pelvic rotation index, pelvic tilt index, or time interval between radiograph and MRI exanimations (Spearman’s rho 0.23, −0.07, and −0.04, respectively), and an additional scatter plot inspection revealed no correlations.

Mean measurements values for all raters in the first round of rating (95% CI) were: OAIR 22.2 degrees (20.2; 24.2), OAIMRI 23.2 degrees (21.2; 25.1), and CAI 10.4 degrees (8.5; 12.4). Mean pelvic tilt index (range) was 0.75 (0.47–1.03), and mean pelvic rotation index (range) was 1.0 (0.72–1.25).

## 4. Discussion

We found a high degree of intrarater reliability and a mediocre interrater reliability in measuring OAI and CAI in pelvic radiographs and MRI scans of children evaluated for borderline AD. The average difference in OAI between MRI and pelvic radiographs was 3.7 degrees, with no trend towards over- or underestimation, with no detectable correlation to pelvic positioning or interval between examinations, although pelvic positioning exceeded recommended limits in 80% of cases.

### 4.1. Interpretation

Pirpiris et al. previously reported on a significant correlation between OAI measurements on pelvic radiographs and MRI scans obtained on the same day, with mean differences of 0.36 degrees [12], but the measurements were made by one rater on a small sample size (seven patients) and did not take positioning of the pelvis on radiographs into account or include mean absolute differences between measurements. If the differences between measurements are not analysed as absolute values, there is a risk of negative and positive differences equalizing each other, resulting in a smaller mean difference, which could mislead the reader into thinking that the difference between measurement methods is smaller than it actually is. The present study expands on these results and show that the mean difference in OAI measurements between radiographs and MRI scans was one degree and the absolute difference was 3.7 degrees when taking pelvic positioning, interval, and multiple raters into account. As the normal range of OAI spans approximately 10 degrees in children [9], a difference in measurements between pelvic radiographs and pelvic MRI of 3.7 degrees could likely contribute to misclassifying a normal hip as borderline dysplastic or vice versa.

Duffy et al. reported high reliability when assessing the OAI on MRI scans [13], but their study required sedation of the children during the MRI scans, in spite of immobilization of the child with a spica cast. We demonstrated that OAI could be measured reliably without the need for sedating the children during the MRI examination.

The high intrarater reliability of the OAI and CAI measurements, found in this study of children evaluated for borderline AD with no previous surgical treatment for hip dysplasia, reflect previous findings when measuring the OAI on pelvic radiographs in children treated conservatively for DDH [14].

Our findings demonstrated that MRI scans can be used to reliably assess the CAI and OAI by a single rater. This indicated that the images were of sufficient quality for measurements; however, the interrater reliability was disappointingly only mediocre. As raters agreed on the image selection for analysis, reasons for this must be sought elsewhere, and it could be speculated that it was not fully reached, even though consensus was sought through a pre-study consensus meeting. This demonstrates the need for standardized measurement guidelines in order to perform MRI measurements consistently across different raters in this rather new image modality for acetabular dysplasia.

The pelvic tilt index was lower than the acceptable values of 0.9–1.4 [15] in 80% of cases in the present study. Despite this prevalence of pelvic mispositioning, we were not able to detect a significant correlation between pelvic positioning and mean absolute difference between OAIR and OAIMRI. Furthermore, while previous studies used precise manipulation of a 3D scan construct to achieve pelvic angulation and tilt, the present paper relied on the tilt/rotation indices calculated from radiograph measurements. It may be that these indices did not accurately reflect the true tilt/rotation of the pelvis, which, combined with our small sample size, may explain why we were unable to reproduce the previous reports of the effect of pelvic mispositioning on OAI measurements by ± five degrees [2,16]. While pelvic mispositioning could affect the accuracy of radiographic measurements, this is not the case for pelvic MRI scans, where the examiner is able to control the positioning of the pelvis 3D construct. The precision of MRI OAI measurements is, therefore, not dependent on the positioning of the pelvis but rather on the image selection by the raters, which we have demonstrated to be reliable. As this was a retrospective study, the proportion of radiographs with unacceptable pelvic tilt was a reflection of the reality facing clinicians, where radiographic measurements guiding surgical decision making may be skewed by pelvic mispositioning by up to ± five degrees and should be interpreted while considering the position of the examined pelvis, as Tönnis pointed out in his original paper on the acetabular index [9]. We believe this in itself to be a strong argument for the use of pelvic MRI before treatment decisions are made regarding acetabular dysplasia.

### 4.2. Limitations

As AD is a rare condition, only a small sample size was obtained during our study period, which must be considered when interpreting the correlation analyses, as there is a risk of a type 2 error. The small sample size could also account for the relatively wide confidence intervals of mean measurement values, mean differences, and the reliability analysis results.

The included children did not have a pelvic radiograph and MRI scan performed on the same day; rather, a mean interval between examinations of 133 days was found. This interval could allow for changes in the examined anatomical structures, but we were not able to detect a correlation to mean absolute differences between OAIR and OAIMRI and interval between examinations. This could, however, be attributed to a type 2 error.

As all raters were aware that their measurements were observed, this study is subject to observation bias, specifically the Hawthorne effect. To limit this, only the primary author of this study had access to the measurement data, but, as the raters were also the co-authors and conceptualizers of this study, this source of bias was not possible to avoid.

For simplicity in analysis, we excluded MRI scans with differing sequence selection by raters in the image selection agreement analysis. This is a source of selection bias and needs to be considered when interpreting the results of the analysis. However, the analysis revealed that there was near total agreement in image selection for that sequence across all raters when raters picked the same sequence, which they did in 80% of cases.

### 4.3. Generalisability

The 16 children included for this study all had pelvic MRI scans performed without sedation. We achieved this by close collaboration with the radiologists, radiographers, and anesthesiologists at our institution, who thoroughly explained the procedure to the parents as to avoid the need for sedation. We accept that performing MRI scans of children without sedation is challenging, and that this may not be achievable at other institutions without significant multidisciplinary collaboration and effort.

Conclusion: OAI and CAI measurements were found to have a high degree of intrarater reliability and mediocre interrater reliability. There was an absolute difference of 3.7 degrees in OAI when measured on pelvic MRI scans as compared to pelvic radiographs.

## Figures and Tables

**Figure 1 children-10-00758-f001:**
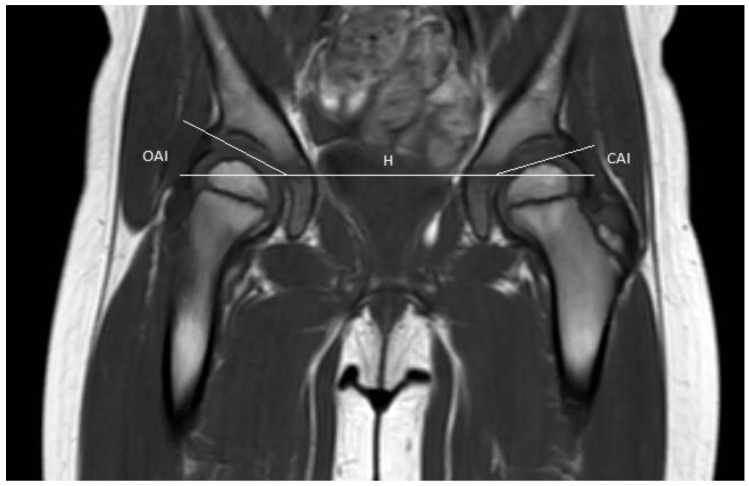
T1 weighted coronal magnetic resonance image scan of a paediatric pelvis. H: Hilgenreiner’s line; OAI: Osseous acetabular index; CAI: cartilaginous acetabular index. CAI was measured as the angle between Hilgenreiner’s line and a line drawn from the lateral margin of the cartilaginous acetabulum at the attachment of the labrum to the superolateral margin of the triradiate cartilage.

**Figure 2 children-10-00758-f002:**
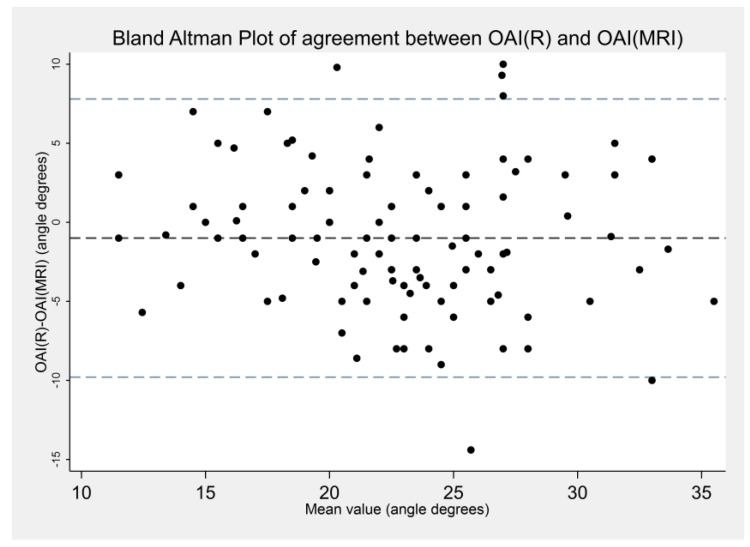
Bland–Altman plot, agreement of OAIR and OAIMRI measurements made by 4 raters in 1 round of rating. Mean difference −0.99, limits of agreement −9.8 to 7.8. OAI = osseous acetabular index.

**Table 1 children-10-00758-t001:** ICC values for inter- and intrarater reliability with 95% bootstrap confidence intervals, bootstrap samples 1000. OAI = osseous acetabular index, CAI = cartilaginous acetabular index.

Measurement	ICC (95% CI)
Interrater reliability	
Osseous, radiographic (OAI_R_)	0.84 (0.65; 0.93)
Osseous, MRI (OAI_MRI_)	0.72 (0.51; 0.84)
Cartilaginous, MRI (CAI)	0.65 (0.44; 0.80)
Intrarater reliability	
Osseous, radiographic (OAI_R_)	0.91 (0.72; 0.98)
Osseous, MRI (OAI_MRI_)	0.84 (0.55; 0.97)
Cartilaginous, MRI (CAI)	0.80 (0.56; 0.95)

**Table 2 children-10-00758-t002:** Mean values, differences, and absolute differences (angle) of bilateral acetabular index measurements made in the first round of rating by all raters. OAI = osseous acetabular index, CAI = cartilaginous acetabular index.

Measurement	Mean Angle (95% CI)
Osseous, radiographic (OAI_R_)	22.2 (20.2; 24.2)
Osseous, MRI (OAI_MRI_)	23.2 (21.2; 25.1)
Cartilaginous, MRI (CAI)	10.4 (8.5; 12.4)
Difference (OAI_R_-OAI_MRI_)	−0.99 (−1.84; −0.16)
Absolute difference (OAI_R_-OAI_MRI_)	3.68 (3.17; 4.20)

## Data Availability

Anonymised data is available upon written request to the main author.

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
