# Peer review of "MRI May Be More Valuable than Pelvic Radiographs in the Assessment of Paediatric Borderline Acetabular Dysplasia"

_children, 2023, doi:10.3390/children10040758_

Round 1

Reviewer 1 Report

Thank you for the opportunity to review this interesting research paper.

In methods - line 72 the authors mention that unacceptable MRI scans were excluded, Then in results - line 149, the authors write that no scans were excluded. How many patients were excluded due to unacceptable imaging? It would be important to know what MRI protocol was used and which scans were exactly performed, including how much time the exam took. It is quite uncommon for 5-year-old patients to be able to tolerate an MRI study without any type of sedation, both due to the length of time required to stay still without motion, as well as the general feeling of enclosed space without parents present. I would like to understand how the authors managed to perform the exams without sedation.

In which position the x-rays were taken? supine or standing? 

The difference in OAI between radiographs and MRI (table 1) is 1 degree. Is it reasonable to perform MRI just because of this minimal difference? Though, the MRI provides also CAI measurement. Though it is mentioned in the introduction the potential clinical use of CAI, it is not mentioned in the discussion. Is there a CAI cutoff to diagnose dysplasia? How the number measured in this paper (average 10 in table 1) is related to the reported numbers in the literature. I would expect to see some mention and discussion of CAI in the discussion section. Especially, in light of the fact that this paper didn't show any advantage of OAI measurement on MRI compared to radiographs.

In the discussion section, the authors mention that radiographic measurements may be skewed by pelvic positioning making the MRI measurements advantageous (lines 253-257). Though the results of this paper failed to show this difference (lines 242-244). Despite that, the authors stick with the conclusion from historic articles. This is misleading. This paragraph should be rephrased to discuss the current conclusion compared to the historic one and discuss the possible reasons for the difference (while mentioned in the limitations paragraph, this should be discussed together and not as a separate entity).

The title of this paper is misleading. There are several conclusions of this paper - that the intrarater reliability is good for all measurements. That the interrater reliability is mediocre stressing the need for a type of consensus on the way to perform the measurements. That the MRI has the potential to provide additional information over standard radiographs though its clinical application is still unclear. Though, the conclusion is not that MRI is more valuable than radiographs. As long as there is no clear clinical benefit that can be proven, it is unreasonable to recommend performing an expensive and stressful imaging modality for small children. 

Author Response

We thank the reviewer for their thorough efforts in reviewing this paper. We have addressed all points below in a point-by-point manner.

In methods - line 72 the authors mention that unacceptable MRI scans were excluded, Then in results - line 149, the authors write that no scans were excluded. How many patients were excluded due to unacceptable imaging?

  • We included the exclusion criteria to clarify for the reader that this qualitative check had been made. We agree with the reviewer that it is unnecessary and have therefore removed the abovementioned sections.

It would be important to know what MRI protocol was used and which scans were exactly performed, including how much time the exam took. It is quite uncommon for 5-year-old patients to be able to tolerate an MRI study without any type of sedation, both due to the length of time required to stay still without motion, as well as the general feeling of enclosed space without parents present. I would like to understand how the authors managed to perform the exams without sedation.

  • We thank the reviewer for this question and agree that MRI of non-sedated children are challenging. We had a close collaboration with the radiologists, the radiographers and the anesthesiologist at our institution, who put a lot of effort into explaining the procedure to the parents and the children to avoid the need for sedation.
  • The measurements were done on a Coronal T1 sequence with a scan time of approx. five minutes which all children tolerated.
  • We have added the information of the scan time to the methods section and added a “generalizability” section explaining how we achieved the scans without sedation.

In which position the x-rays were taken? supine or standing? 

  • All x-rays were performed with the child in the supine position. This information has now been added to the methods section.

The difference in OAI between radiographs and MRI (table 1) is 1 degree. Is it reasonable to perform MRI just because of this minimal difference? Though, the MRI provides also CAI measurement. Though it is mentioned in the introduction the potential clinical use of CAI, it is not mentioned in the discussion. Is there a CAI cutoff to diagnose dysplasia? How the number measured in this paper (average 10 in table 1) is related to the reported numbers in the literature. I would expect to see some mention and discussion of CAI in the discussion section. Especially, in light of the fact that this paper didn't show any advantage of OAI measurement on MRI compared to radiographs.

  • A very good point which we have not communicated clearly. The addition of the relative difference is and Bland-Altman plot illustrates that there is no systematic trend towards over- or underestimation of OAI between imaging modalities and that this difference is not dependent on the mean OAI measurements. As the absolute differences indicate, the difference between MRI and XR (or disagreement) is 3.7 degrees, which “could likely contribute to misclassifying a normal hip as borderline dysplastic or vice versa” line 223.
  • Due to manuscript restrictions we were unable to accommodate a discussion of the CAI and chose to focus on the OAI findings, as they are more widely used.

We thank the reviewer for pointing this out and have edited the manuscript to reflect this point more clearly.

In the discussion section, the authors mention that radiographic measurements may be skewed by pelvic positioning making the MRI measurements advantageous (lines 253-257). Though the results of this paper failed to show this difference (lines 242-244). Despite that, the authors stick with the conclusion from historic articles. This is misleading. This paragraph should be rephrased to discuss the current conclusion compared to the historic one and discuss the possible reasons for the difference (while mentioned in the limitations paragraph, this should be discussed together and not as a separate entity).

  • Thank you for pointing out this apparent discrepancy. In the studies by Hamano and Van der bom (reference 2 and 16) multiple radiographs were constructed from CT and MRI thus enabling total control of the pelvic rotation and tilt. In the present study the sample size was relatively small and relied instead on the pelvic tilt and rotation indices. We speculate that these indices based on radiographic ratios do not accurately reflect the true tilt/rotation of the pelvis, which, combined with our small sample size, may explain why we are unable to reproduce the results from reference study 2 and 16. We therefore chose to report our findings while still considering the historical articles valid.
  • The manuscript has now been edited to reflect this point

The title of this paper is misleading. There are several conclusions of this paper - that the intrarater reliability is good for all measurements. That the interrater reliability is mediocre stressing the need for a type of consensus on the way to perform the measurements. That the MRI has the potential to provide additional information over standard radiographs though its clinical application is still unclear.

Though, the conclusion is not that MRI is more valuable than radiographs. As long as there is no clear clinical benefit that can be proven, it is unreasonable to recommend performing an expensive and stressful imaging modality for small children. 

  • We agree, which also is not the conclusion of the paper. In the title we state that MRI may be more valuable as we detect mean absolute differences across all raters in OAI between modalities of up to 3.7 degrees, which may convert the classification of a normal hip to abnormal or vice versa.
  • Further, it adds the additional information of the CAI, which, while we did not discuss this, is presented in the introduction as having potentially prognostic value for RAD.

Reviewer 2 Report

1. In the manuscript titled “MRI may be more valuable than pelvic radiographs in the assessment of paediatric borderline acetabular dysplasia” the authors retrospectively evaluated the reliability of two indexes used for acetabular dysplasia diagnosis in children, comparing radiography and magnetic resonance imaging.
2. The topic may be considered relevant to the field, as magnetic resonance imaging has been shown to be more accurate for treatment selection (10.1097/BPO.0b013e3181c877d7), but its reliability has not been studied.
3. The manuscript shows that two imaging modalities have a high intrarater reliability but mediocre interrater reliability. With a relatively high absolute difference of 3.7 degrees for one index between two modalities, the result highlights the need to use a select image modality.
4. The methodology requires clarification on the part of study design. To the best of my knowledge, magnetic resonance imaging is not routinely used for acetabular dysplasia assessment, requiring additional Ethical Committee work-up. Furthermore, if a non-standard scan protocol was used, it would require an informed consent form. Please clarify. Consider also providing suggestions on increasing interrater correlation for use in clinical practice.
5. The conclusions are consistent with the evidence and arguments presented.
6. The references are appropriate.
7. The reviewer has no additional comments on the tables and figures.

Author Response

We thank the reviewer for their thorough and valuable effort in reviewing this paper. Below we have answered the points raised in a point-by-point manner.

In the manuscript titled “MRI may be more valuable than pelvic radiographs in the assessment of paediatric borderline acetabular dysplasia” the authors retrospectively evaluated the reliability of two indexes used for acetabular dysplasia diagnosis in children, comparing radiography and magnetic resonance imaging.
2. The topic may be considered relevant to the field, as magnetic resonance imaging has been shown to be more accurate for treatment selection (10.1097/BPO.0b013e3181c877d7), but its reliability has not been studied.
3. The manuscript shows that two imaging modalities have a high intrarater reliability but mediocre interrater reliability. With a relatively high absolute difference of 3.7 degrees for one index between two modalities, the result highlights the need to use a select image modality.
4. The methodology requires clarification on the part of study design. To the best of my knowledge, magnetic resonance imaging is not routinely used for acetabular dysplasia assessment, requiring additional Ethical Committee work-up. Furthermore, if a non-standard scan protocol was used, it would require an informed consent form. Please clarify. Consider also providing suggestions on increasing interrater correlation for use in clinical practice.

  • Pelvic MRI was part of the diagnostic routine for RAD at our institution during this period. As this protocol was not part of any clinical research project but rather a retrospective analysis of MRI scans performed during routine diagnostics, ethical approval was not needed.

  1. The conclusions are consistent with the evidence and arguments presented.
    6. The references are appropriate.
    7. The reviewer has no additional comments on the tables and figures.

Round 2

Reviewer 1 Report

Thanks for the opportunity to review the paper again. 

I'm happy with the new version and recommend publishing in current form.

Reviewer 2 Report

The authors have answered the reviewer's questions. The provided manuscript version is further improved.